# The Hybrid Stylus: A Multi-Surface Active Stylus for Interacting with and Handwriting on Paper, Tabletop Display or Both

**DOI:** 10.3390/s22187058

**Published:** 2022-09-18

**Authors:** Cuauhtli Campos, Jakub Sandak, Matjaž Kljun, Klen Čopič Pucihar

**Affiliations:** 1Faculty of Mathematics, Natural Sciences and Information Technology (FAMNIT), University of Primorska, 6000 Koper, Slovenia; 2InnoRenew CoE, 6310 Isola, Slovenia; 3Faculty of Information Studies, 8000 Novo Mesto, Slovenia

**Keywords:** pen-paper user interfaces, digitisation, handwriting, tablet computers, tabletop computers

## Abstract

The distinct properties and affordances of paper provide benefits that enabled paper to maintain an important role in the digital age. This is so much so, that some pen–paper interaction has been imitated in the digital world with touchscreens and stylus pens. Because digital medium also provides several advantages not available to physical paper, there is a clear benefit to merge the two mediums. Despite the plethora of concepts, prototypes and systems to digitise handwritten information on paper, these systems require specially prepared paper, complex setups and software, which can be used solely in combination with paper, and, most importantly, do not support the concurrent precise interaction with both mediums (paper and touchscreen) using one pen only. In this paper, we present the design, fabrication and evaluation of the *Hybrid Stylus.* The *Hybrid Stylus* is assembled with the infinity pencil tip (nib) made of graphite and a specially designed shielded tip holder that is attached to an active stylus. The stylus can be used for writing on a physical paper, while it still maintains all the features needed for tablet interaction. Moreover, the stylus also allows simultaneous digitisation of handwritten information on the paper when the paper is placed on the tablet screen. In order to evaluate the concept, we also add a user-friendly manual alignment of paper position on the underlying tablet computer The evaluation demonstrates that the system achieves almost perfect digitisation of strokes (98.6% of strokes were correctly registered with only 1.2% of ghost strokes) whilst maintaining excellent user experience of writing with a pencil on the paper.

## 1. Introduction

Digital mediums offer several advantages over paper. For instance, digital documents can easily be searched, updated, copied, immediately shared over the internet, and linked to other documents and multimedia content. However, despite the omnipresence of digital screens, the paperless office myth [1] has still not prevailed to this day. According to Steimle [2], the prevalence of paper cannot be merely attributed to the shortcomings of current display technologies, such as limited screen size, resolution, contrast, dependency on power, etc. Instead, as suggested by a large body of research [3,4,5,6,7,8,9], paper supports a wealth of interactions that have a number of inherent advantages over digital technologies [2]. For instance, paper carries unique properties such as tangibility (e.g., making information tangible), accessibility and simplicity of use (e.g., annotating on it is easy, flexible and is smoothly integrated with reading [2]), versatility (it can be easily reshaped and folded allowing its use in different contexts), and endurance (it can be preserved over thousands of years). Furthermore, it also does not require power to operate and specific software to show stored information, which is an important advantage when considering mobility and long-term preservation. Our society thus remains heavily reliant on paper.

The emergence of digital technologies has brought forward the idea to combine paper with digital mediums. It is thus not surprising that numerous concepts to merge the richness of paper interactions with the flexibility of digital content have been proposed, developed and explored [10]. A common coupling involves displaying digital information and commands (buttons, knobs and sliders) above the paper [11,12,13,14], on [15,16,17], besides [3,18] or beneath the paper [19,20,21,22]. Digital content can be displayed by either projecting it on and around paper, or by showing it on a nearby display.

Another possibility to couple both mediums, while preserving pen–paper interaction, is by capturing and digitising handwritten content on a physical paper. To support such an interaction, pen strokes on the paper need to be tracked, for example, (i) by tracking hidden patterns (pre-printed) on a paper with a special pen (e.g., Anoto digital pen and paper (Anoto pencil: https://www.anoto.com/ (accessed on 31 August 2022))), (ii) by tracking a regular pen captured with an RGB camera and processed by an image-recognition system [23,24,25], or by (iii) a combined analysis of sound emitted by a stroking pen and data captured by a pressure sensor, gyroscope and accelerometers [26]. However, such systems are either limited to a predefined set of gestures, require specially prepared paper (e.g., with a pre-printed hidden pattern) or complex setups and software, while dedicated pens required for such interaction can still be used solely with paper.

In the last decade, tablet computers, graphic tablets with LCD displays and tabletop computers (also table PC) have also tried to imitate writing and drawing on paper with stylus pens and touchscreens. However, to the best of our knowledge, no system exists that would enable switching between all three use contexts: digital context (i.e., interacting with a tablet computer only), physical context (i.e., drawing and writing on paper), and combined context (i.e., interacting with tablet and paper simultaneously). The advantages of switching between the contexts can already be used on the tablet computer, as can be observed in Figure 1, where the same stylus can be used for writing on the paper, interacting with the touchscreen and digitising handwritten content. With further proliferation of the aforementioned large horizontal touchscreens, we can also envision the future where physical desktops will be made of such displays with paper documents scattered on top (Figure 1 right). Interaction for all three contexts on such displays is evidently beneficial and this is where the motivation for our research spans from.

Supporting interaction within all three contexts opens up several research questions. In this paper, we explored the following. How to design a hybrid stylus tip that will support handwriting on a paper and still provide interactive capabilities of an active stylus on a tablet computer? How to maintain interactive capabilities of the stylus even when the tablet is covered with a sheet of paper? How to align paper on the underlying tablet computer in order to (i) link physical documents with digital counterparts to correctly digitise pen strokes on the latter, and (ii) have a demonstrable prototype of a hybrid stylus? To this end we propose, design and evaluate a hybrid stylus by modifying an existing active stylus and coupling it with a graphite tip that supports handwriting on paper as well as interaction with commonly available off-the-shelf tablet computers (e.g., iPad Pro, Samsung Galaxy Tab S4). Moreover, the system supports digitising pen strokes by placing a sheet of paper on the screen and aligning its position, which enables writing on paper, tablet or both simultaneously. This paper provides the following contributions:The design and fabrication process of the *Hybrid Stylus* by modifying an off-the-shelf active stylus.The results of a user study (n=12) evaluating interaction techniques (selected in elicitation study (n=6)) for manual alignment of paper position on the underlying tablet computer, needed for evaluating the stylus and demonstrating real use-cases.The results of a user study (n=12) performing qualitative and quantitative evaluation of our *Hybrid Stylus* specifically designed to evaluate user experience and stroke digitisation accuracy.Presentation of several prototypes demonstrating real world use-cases for the *Hybrid Stylus*.

The next section presents the related work in the field (Section 2) followed by the description of materials and methods (Section 3), and the description of three user studies and their results (Section 4). The paper concludes with a presentation of practical applications of our system (Section 5) and discussion (Section 6).

## 2. Background and Related Work

The following subsections present different technologies used in computer stylus pens to enable pen computing interfaces, approaches to digitise pen strokes made on paper, and approaches to track paper or other tangible objects in order to couple them with digital counterparts.

### 2.1. Computer Stylus Pens (or Styli)

Different types of computer stylus pens (or styli) exist that support pen computing interfaces [27]. The first types are the *passive styli*, which include a capacitive-type touch system (CTS) styli [28,29,30], and digital pens that further include Electromagnetic Resonance (EMR) styli (e.g., Samsung Galaxy Note 10 S-Pen (Samsung S-Pen: https://www.samsung.com/us/mobile/mobile-accessories/phones/galaxy-note10-s-pen-black-ej-pn970bbegus/ (accessed on 31 August 2022))) [31,32] and Electrically Coupled Resonance (ECR) styli [32]. The second types are the *active styli*, also known as CTS active styli [30] (e.g., Apple Pencil (Apple pencil: https://support.apple.com/kb/SP786 (accessed on 31 August 2022))) or Microsoft Surface Pen (Microsoft Surface Pen: https://www.microsoft.com/en-us/d/surface-pen/92fp8q09qhxc?activetab=pivot:overviewtab (accessed on 31 August 2022)).

The main differences between the two types include the tracking method used and the battery integration. The CTS *passive styli* use electrostatic conduction, just like fingers on the touchscreen, and do not require any electronic component or battery. *Active styli* feature a digitising sensor on the pen tip that communicates with the display, whereas *digital pens* use EMR or ECR to communicate with an interactive display (e.g., Wacom creative pen displays (Wacom: https://www.wacom.com/en-ch/for-business/technologies/emr (accessed on 31 August 2022))) [33]. Besides tracking the location of a pen touching the screen, some devices are also capable of tracking the pressure and orientation (tilt) of the tip. For example, the Apple pencil is using a multiaxial force sensor, which enables three dimensional force estimation and is important for high fidelity stroke digitisation [34]. Such a feature is not available in C-type *passive styli* that use a larger and rounded tip for robust detection [27].

Due to the bulky tip of the CTS *passive styli* it is difficult to perform precise actions, while higher latency makes these pens less responsive compared to other technologies described. However, as such devices do not feature any electronics, they do not require battery power. Since *digital pens* work on the principle of EMR or ECR, they also do not require battery power. These pens feature inbuilt electronics charged through the magnetic field of a digitiser integrated in a touch-enabled display.

While the solutions presented are designed to enable users to enter information into digital electronic devices by writing on a screen, they are not capable of writing or leaving notable traces on paper. An open research question we explore in this paper is how to modify a digital pen so that it can be used both to write on paper as well as capture pen strokes on a digital device. The writing should be possible on both mediums at the same time when a paper is placed on a digital device, or individually on each medium separately. Existing solutions for digitising pen strokes on paper and their drawbacks are presented in the next subsection.

### 2.2. Digitising Strokes on Paper

Early solutions to digitise handwritten information on paper used either handheld scanners, cameras to capture video/pictures of handwritten information, or track pen movements. For example, ARIEL [35] and Video Mosaic [36] allowed users to scan or capture paper as well as project digital information over physical paper with a projection-based augmented reality (AR). DigitalDesk [23] and FACT [24] also featured cameras for capturing AR content of the digital projection together with paper documents on a desk and image recognition system in order to track pen movements and “read” content on paper. Similar systems called Mercel [37] and CamWorks [38] used a camera to capture physical paper together with an OCR software to digitise handwritten information that could be displayed on a digital document on a nearby computer display. Another approach proposed by Grbac et al. [39] was to use a mobile phone camera to digitise text highlights in close to real time and show them on an e-document displayed on a laptop computer. A more recent outside-in camera based digitisation system that offers sub-millimeter accuracy was proposed by Wu et al. [40]. All these solutions either require time to manually scan documents, require multiple devices (cameras, image recognition system, OCR and other software) to operate and are affected by cameras’ resolution and lightning conditions in users’ surroundings. In addition, users’ hands can affect projection and image recognition systems while covering content or distorting projection on the desk.

Another method to digitise pen strokes is to use an integrated camera into a pen. One of the earliest systems was PaperLink [25] that used a VideoPen prototype with a small mounted camera, which worked in a similar way as the systems described above. A commercial system using just a camera called eBeam was also released (eBeam smartpen: https://www.indiegogo.com/projects/ebeam-smartpen-the-last-pen-you-ll-ever-need#/ (accessed on 31 August 2022)). More sophisticated are the pens with a camera that tracks pre-printed hidden patterns on the paper. Examples of such pens include Neo smartpens (Neo smartpens: https://neosmartpen.com/product-lamy/ (accessed on 31 August 2022) and Anoto (Anoto pencil: https://www.anoto.com/ (accessed on 31 August 2022), Livescribe: https://eu.livescribe.com/ (accessed on 31 August 2022)). Several prototypes tried to expand the capabilities of such pens.

Researchers have, for example, enabled paper-based documents to be transferred to and further edited on a digital whiteboard [41]. Some prototypes expanded pen capabilities with additional (gesture-based) commands by using another camera [14,42] or tabletop display [43] to capture finger and pen gestures. These gestures could initialise different actions such as linking documents, display menus, capture handwriting, etc. The CoScribe [43] and NiCE [44] systems also allowed users to collaboratively annotate, link and tag both printed and digital documents using physical documents as a proxy and visualising digitised information on a tabletop or whiteboard.

The PADD system allowed any digital document to be printed and annotated with a camera-based pen and digitised back to the original document [45]. PaperProof expanded this idea by converting pen operations and handwriting on a printed document to the tracking changes functionality of a word processor on a computer [46]. Meanwhile, the Penbook used a tablet computer with a projector projecting digital information on a paper that could be coupled by handwritten information with a pen [47]. It has to be noted that all these solutions use a specially designed pen and paper as well as additional cameras in order to transfer handwritten information from the paper to a digital form on a device that is not necessarily under the paper. However, to the best of our knowledge, such pens are not capable of writing directly on the screen without physically replacing the ink-cartridge tip with a stylus tip.

### 2.3. Tracking and Identifying Tangible Objects

In this section, we look at how objects such as paper documents on the desktop can be tracked and identified. For a working prototype, the computer needs to “know” where the document is and which document users are working on in order to successfully couple (align, merge) the paper document to its digital counterpart and to correctly digitise stylus strokes. Identification and tracking or alignment are thus an integral part of a hybrid stylus concept and are reviewed hereafter.

Tracking of physical documents can be accomplished with a camera or multiple cameras, as used in the previous section. In order to “recognise” and track physical documents with a camera, special codes can be printed on the margins of the paper such as in PaperSpace [48] and DocuDesk [22]. These computer vision-based systems allowed users to locate paper copies of printed digital documents and retrieve their digital versions based on these codes. In addition, in the PaperSpace system, a set of instructions (annotate, open, link, email, information) could be printed in the margins of a physical document as well, which could be activated by system-recognised gestures on them (e.g., by selecting an email icon on the margin of a physical document resulted in attached digital counterpart to an email). Similar commands were demonstrated by the paper on the tabletop in the DocuDesk system.

Another possibility for identifying and tracking objects is to use electronic markers such as RFID tags or dedicated trackers such as the HTC Vive tracker (HTC Vive tracker https://www.vive.com/us/accessory/tracker3/ (accessed on 31 August 2022)). Such a system offers high quality tracking with high accuracy and low latency; however, due to the electrical components used, they are bulky and relatively high priced compared to the camera -based solutions.

An alternative to track physical objects are Passive Untouched Capacitive Widgets (PUCs) [49]. PUCs are devices that provide touch-like behaviours on capacitive multi-touch displays. In essence, they function as an electrical conductor between the user and the touch surface by grounding the widget marker and can be distinguished by using various sizes of rounded pads. Persistently Trackable Tangibles on Capacitive Multi-Touch Displays (PERCs) are an evolution of PUCs that overcome the limitation PUCs by (i) adding the system to determine which tangibles are currently placed on an interactive surface and whether they are being touched or not, (ii) making each PERC unique, and (iii) adding the ability to detect the exact position and orientation of each tangible. These enhancements were achieved by adding a field sensor, a light sensor, a micro controller and a Bluetooth element into a PUC widget [50]. Both camera-based computer vision systems and PUCs/PERCs require dedicated hardware and complex setups to achieve tracking.

For evaluating and demonstrating the *Hybrid Stylus* concept, these tracking and identification techniques were too complex. We decided to design and evaluate a simple and user-friendly manual alignment of paper position on the underlying tablet computer. For the purpose of the concept, identification of documents could also be done manually (selected by clicking on the appropriate button on a touchscreen). One idea to align the paper is to use simple stroke actions over the paper using a stylus. Existing hand gesture classifications are presented in the next subsection.

### 2.4. Hand Gesture Classification

Capacitive and resistive touchscreens are commonly used for detecting gestures and can enable a variety of interactions from issuing commands to software [51] and 3D object manipulation [52], to physical therapy [53,54], to name a few. There are also prototypes that enable designers to define and evaluate their own newly created gestures [55].

Gestures are commonly captured as a time-series of 2D points detected on a touchscreen surface. These data are then processed by various gesture recognisers such as G-Gene [56], !FTL [57], $Q [58], and $P [59]. These recognisers rely on the historical Rotate-Scale-Translate (RST) paradigm [60] that is still a common method used to date. The paradigms uses the parameters captured from two or more reference points to ensure the invariance of the “rotation (relying on angular data), scaling (based on length ratio), and translation (linked to the barycenter of the translation)” [61], which is needed for a captured gesture to be successfully compared to a reference gesture.

However, the RTS paradigm is not convenient when it comes to more complex gesture-based operations, such as editing, cutting, copying, pasting, deforming, and composing gestures [62]. Researchers have introduced other model-based approaches to address this issue such as Bézier curves and surfaces [63] or B-Splines [62] and others. However, these approaches are resource consuming and complex to understand, design, and develop. Recent approaches use, for example, Bézier curves to represent, store, and manipulate touch-surface multi-stroke gestures, but in addition, provide a higher level of abstraction, as used in the RTS paradigm [61].

Machine learning algorithms are also commonly used for handwriting recognition even when a low amount of pixels is available [64] or for authenticating users [65]. However, such an approach usually requires large datasets. Nevertheless, machine learning algorithms enabling lightweight computation also exists [53]. This limitation can be also mitigated through the use of kinematic theory, which is capable of generating synthetic human-like gestures [66,67]. Another approach is to utilise different input signals such as electromyography-based gesture recognition systems [54].

Using Bézier curves or B-Splines, one can create accurate stroke descriptors that can be used to create a simple gesture detection system. This approach was followed in this research for the implementation of the manual alignment of paper position on the tablet screen.

### 2.5. Unexplored Areas

To the best of our knowledge, a hybrid stylus capable of writing on and interacting with paper and touchscreen at the same time does not exist. We thus decided to investigate the possibility of developing an active stylus pen that could be used for writing on paper while preserving the richness of interaction of an active stylus pen with touchscreens. In addition, the writing should be possible on both mediums at the same time when a paper is placed on a digital device.

In order to evaluate a hybrid stylus and demonstrate its use-cases, we had to explore how to align paper with its digital counterpart on a touch screen when a paper is placed on the tabletop or tablet computers in order to capture handwriting on both mediums. As mentioned above, several systems supporting paper tracking exist using cameras or specially developed electronics. However, we wanted to explore a possibility to build such a tool using off-the-shelf hardware and modify it in such a way that any person could replicate it.

## 3. Materials and Methods

We designed and fabricated a hybrid stylus capable of writing and drawing both on the paper, tablet screen or both at the same time. In order to evaluate the feasibility of the proposed system, we conducted three user studies. The first one was the elicitation study for selecting preferable stroke action for the alignment of paper position on a tablet screen. The other two user studies Study A and B evaluated (i) stroke actions that were selected in the elicitation study, and (ii) the *Hybrid Stylus* usability. The following subsections present the design and fabrication of the *Hybrid Stylus*, apparatus used, participant sampling, and statistical methods.

### 3.1. The Design and Fabrication of the Hybrid Stylus

The overall objective was to fabricate the tip of an active stylus pen so that it would be capable of writing on paper, tablet screen, or simultaneously on both mediums, while still preserving all properties of an active pen while interacting with the tablet. Moreover, we aimed to deliver a low cost long-lasting drawing and writing solution that would be based on the existing active stylus pen, such as Apple Pencil or Microsoft Surface Pen.

The *Hybrid Stylus* was designed through an iterative process, taking the 2nd generation Apple Pencil as the basis. Using SolidWorks Computer Aided Design (CAD) software, we designed, modelled, and 3D printed several different adapters for mounting an alternative (pen) tip on the stylus. We converged to two adapter designs. The first one was a double pen adapter designed to hold an additional pen in close proximity to the stylus (Figure 2a). This design required writing on paper using two contact points, which raised some usability issues. The second was a hybrid tip design, in which we extended the tip of the stylus with extension tips (Figure 2b) fabricated from various materials.

We experimented with pen tips made of classical pencil lead, ball pen nib, marker nib, and infinite pencil tip (Ainiv Everlasting Pencil is made of a graphite material similar to https://theinfinitypencil.com/ (accessed on 31 August 2022)). We identified that the infinite pencil tip made from a proprietary metal alloy (usually consisting of lead, silver, copper, and/or gold) was the most suitable due to its favourable electromagnetic properties (e.g., high conductivity), as well as relatively high hardness and elevated wear resistance. These properties enable an extended lifetime of the tip and offer a good writing and drawing experience on various materials, such as paper, similarly to pencil. Furthermore, the high hardness of the extension tip allowed us to use the pencil directly on the tablet without leaving a trace on the screen. The experimentation also revealed that the good electrical connection between the stylus tip and the infinite pencil tip must be assured (marked in Figure 2b). To achieve this, we designed and 3D printed an adapter that has a tight fit with the smart pen. The extension tip is then screwed into the adapter, forming a tight contact ensuring good electrical connection.

The mechanical arrangement of the inserted extension tip did not affect the force sensing capacity, particularly in the axial direction. However, the modification of the original smart pencil’s geometrical configuration altered its capacitance profile as well as other dielectric characteristics of the mutual capacitive touch system (touchscreen-stylus couple). As the relative positions of the tip and ring electrodes (measured from the surface of the touchscreen) increased, such a set-up resulted in the improper sensing of the stylus position and tilt, as well as in the generation of random ghost stroke events. This was a result of the improper interpretation of acquired signals by the controlling software due to distorted capacitance variations between electrodes after stylus modification [27]. It is possible to reconfigure the software setting to compensate for altered capacitance fingerprinting; however, such a solution is highly problematic, as it requires low level driver access, which is commonly not available in off-the-shelf stylus systems. Proceeding using this approach would thus greatly limit the broad adaptation of our solution. Conversely, we tried to mitigate this problem by introducing an aluminium foil shield placed around the infinite pen tip. This resulted in a highly effective electrical shielding when the metal was in contact with the users’ skin (finger tips holding the stylus). In that case, the capacitance profile of the *Hybrid Stylus* matched the original device characteristics, thus enabling the (optimal) stylus operation. It should also be mentioned that the optimal performance was noticed when the distance of the aluminium shield to the touchscreen was between 6 and 9 mm, assuming a 12 mm diameter of the shield.

The final configuration of the *Hybrid Stylus* is presented in Figure 2b, as well as the video figure that is part of Appendix A. The informal experimentation revealed that the handling sensation/experience was nearly identical as with the not-enhanced smart pen. Hand or palm resting was not a problem, as the iPad features an excellent palm filtering system already, which worked without problems even when the paper was placed on top of the tablet computer. The new tip did also not create ghost marks on the tablet computer screen when hovering with the pen at various heights. The comparison of interaction techniques can be observed in Table 1. All interaction techniques were comparable between both styli, except for the pressure sensitivity that supports, for example, changing the thickness of the stroke while drawing. Since this feature was not needed for our experiments, the current capabilities of the stylus allowed us to proceed with a formal evaluation of the concept.

### 3.2. Apparatus

Our testing rig (Figure 3) consisted of a desk, chair, tablet computer, laptop computer, and a motion capture system with five infrared cameras (Optitrack Prime 13W cameras were used: https://optitrack.com/cameras/prime-13w/ (accessed on 31 August 2022)). The tablet computer used was an iPad Pro 3rd generation with the screen resolution 2732 × 2048 pixels, pixel density of 264 pixels-per-inch, and iOS operating system. It was placed on the desk at a predefined position that was aligned with the origin and coordinates of the motion capture system.

In Studies A and B, we placed a Copying Paper Papyrus of 80 g/m${}^{2}$ on the tablet computer to complete the tasks with the *Hybrid Stylus*. The tasks are explained in Section 4.2 and Section 4.3. All software was developed using the PencilKit framework (PencilKit documentation: https://developer.apple.com/documentation/pencilkit (accessed on 31 August 2022)) in Apple Xcode (Swift 5, iOS 15.4 SDK) and deployed to an iPad Pro 3rd generation with the 2nd generation Apple Pencil. For detecting stroke actions (e.g., to align paper position) we used the PencilKit’s cubic B-spline control points (PKStrokePath documentation: https://developer.apple.com/documentation/pencilkit/pkstrokepath (accessed on 31 August 2022)).

### 3.3. Participants

We recruited students and employees from the university to participate in the study using convenience sampling. In the elicitation study, 6 participants participated (20–44 years old, 2 females). For Studies A and B, we recruited 12 participants (18–39 years old, 4 females). None of the participants had any precondition that would affect their ability of handwriting or drawing and all but one participant were right-handed. All participants demonstrated interest in using the system for digitising handwriting on a physical paper and said to have experience using mobile phones and tables.

### 3.4. Statistical Analysis

Unless specified differently, the statistical analysis of Study A was carried out using the paired sample *t*-test. Calculations were conducted using base functions in R software on aggregated data. We indicate effect sizes whenever feasible and report our results by stating *p* values and 95% confidence intervals. Data that did not follow a normal distribution (Shapiro-Wilk normality test) were log-transformed for positive skewness. In the case of Study B, we make a descriptive analysis of collected data reporting ratios.

## 4. User Studies

In the following subsections, we describe the three studies conducted and present the results. The first is the elicitation study to select the preferable stroke action for the alignment of paper position on a tablet screen. Study A was conducted to perform qualitative and quantitative evaluation of stroke actions, whilst Study B was performed to evaluate the usability of the *Hybrid Stylus* for writing and drawing on the paper and tablet computer.

### 4.1. Elicitation Study: Selecting Preferable Stroke Actions for Alignment of Paper Position on a Tablet Screen

To successfully digitise information written on a physical paper, the paper position needs to be known in order to overlap it with the digital counterpart on the tablet screen. To align the position of the paper on a tablet screen, we designed a user-friendly interaction technique based on two target markers printed on the paper on which participants would perform a desired stroke action using our *Hybrid Stylus*. This action would generate two point correspondences between the paper and a tablet screen, providing sufficient information for calculating the exact paper position.

Participants were presented with a worksheet placed on the top of the tablet screen and asked what gesture they would most likely make with a stylus to instruct the system of the exact location of the paper. We repeated the question two more times by asking for the second and third most likely gesture, which resulted in 18 responses (6 participants × 3 trials). As can be observed in Figure 4a, 66% of participants used crosses as preferred stroke action, followed by 33% of participants who used dots. Interestingly, no other stroke was used as a first selection. When aggregating all responses (Figure 4b) the cross and the dot were each preferred in nearly a third of all cases, whereas other stroke actions, such as stars, squares, circles and lines were attributed to smaller percentages. For this reason, we used the dot and cross stroke actions as two strokes over target markers for Study A, as presented in the following section.

### 4.2. Study A: Evaluation of Stroke Actions for Alignment of Paper Position on a Tablet Screen

In this study, we evaluated whether it is possible to align paper position on the tablet screen with enough accuracy. We used two A5 size papers, one for each condition (see Figure 5a). On each paper, we placed a constellation of four IR-markers to precisely track the position of the paper using infrared cameras. In one condition, the paper had two target markers (TM1 & TM2) with a visual hint in the form of a cross, and participants were asked to perform a cross gesture over target markers with the *Hybrid Stylus* to align the paper position. In the other condition, the two target markers had a visual hint in the form of a dot and participants were asked to perform a dot gesture (tap on the dot) to align paper position. The cross and the dot were presented in red as a visual hint participants had to follow. The visual hint also included the red dot of r=0.56 mm printed in the middle, hereafter called the *Spawning Anchor Printed Reference* (SAPR).

In both conditions, participants first performed the stroke starting with the left target marker (TM1) followed by the right target marker (TM2). They were also asked to follow the red visual hint on the target marker as precisely as possible, making sure the *Hybrid Stylus* passed throughout the centre over SAPR. Each participant completed 80 stroke actions (20 [trials] × 2 [strokes] × 2 [target markers]). The trail of the graphite left by the *Hybrid Stylus* was erased after every fifth repetition. We used new papers for each user. In order to balance the experiment, we alternated the order of stroke action types. Lastly, participants were asked to choose the preferred stroke action.

The variables measured in Study A were:*Task completion time*. Defined as the time taken by each participant to complete each trial.*Position error*. The distance between SAPR on the paper (measured with the motion capture system) and the centroid of the stroke action on a tablet screen. In case of the dot stroke actions, we used the initial touch point as the centroid (i.e., the first point detected as the stylus touched the paper/screen). For the cross gesture, we calculated the intersection of two strokes (i.e., the strokes were generated as the stylus touched the paper/screen) and if no intersection was found, participants were required to repeat the procedure.*Angle error*. The difference between the angle of paper rotation measured by the motion capture system and the angle calculated from the centroids of two stroke actions (on TM1 and TM2) performed on a tablet screen.

#### Results

All participants managed to complete the study successfully and the vast majority (75%) preferred the dot stroke action as seen in Figure 6e. The homogeneity of variance (Levene) failed for the *Task completion time* variable; therefore, we performed a paired *t*-test of unequal variances. The results demonstrated a significant effect of stroke action ($t\left(11\right)=5.48,\;p<0.001,\;Cohen^{\prime }s\;d=0.45,\;conf.\;int.=\left[1.05,2.46\right]$), where the dot outperformed the cross stroke action (Figure 6a).

When analysing the *Positioning error* of TM1, a significant effect of the stroke action was found ($t\left(11\right)=2.26,\;p=0.044,\;Cohen^{\prime }s\;d=0.8,\;conf.\;int.=\left[0.007,0.536\right]$). The dot stroke action again performed better (x=0.51; Figure 6b). No significant effect was found when analysing accuracy of TM2 ($t\left(11\right)=1.13,\;p=0.28,\;Cohen^{\prime }s\;d=2.0,\;conf.\;int.=\left[-0.102,0.319\right]$). No significant effect was also found for the *Angle error* ($t\left(11\right)=1.91,\;p=0.08,\;Cohen^{\prime }s\;d=0.19,\;conf.\;int.=\left[-0.011,0.167\right]$) Figure 6c).

### 4.3. Study B: Evaluation of the Hybrid Stylus Usability for Writing and Drawing on Paper

In this study, we conducted a qualitative and quantitative evaluation of the *Hybrid Stylus* focusing on simultaneous drawing and handwriting on paper and tablet. A worksheet paper of 190 mm × 230 mm in size with no IR-markers (Figure 5b) was placed on top of the tablet screen and participants were asked to fill it in using the *Hybrid Stylus*. The instructions for the corresponding task were printed on the paper. Participants were first asked to draw three different shapes (i.e., a circle, a square, and a triangle) within the designated areas on the worksheet. This was repeated three times. Afterwards, participants were asked to sign their name and write it in small and capital letters. This was again repeated three times. While drawing using the *Hybrid Stylus*, participants were instructed to hold it in a way that they touch the aluminium shield. If they held the stylus differently, the experimenter reminded them how to hold the stylus. After completing the worksheet, participants were shown a digital counterpart of their worksheet on the tablet screen. After inspecting it, they were asked the following questions: (i) Q1: Was it hard to write, sign, or draw on paper with the *Hybrid Stylus*?; (ii) Q2: Are you satisfied with the digital counterpart of your writing, signature, and drawing? They could respond by selecting one of the three options: *Yes*, *No*, and *I do not know*.

After participants finished both Study A and Study B, they were asked to complete two standard questionnaires: (i) the System Usability Scale (SUS) [68] and (ii) the longer version of the User Experience Questionnaire (UEQ) [69,70]. Both questionnaires are common tools to measure the usability and user experience (e.g., [71,72]). Lastly, participants were asked an open question about their general opinion about the system.

#### Results

All but one participant managed to complete the study successfully. The pen did not work properly for one participant as she accidentally switched the pen mode to eraser. We only report data of 11 participants in this section.

*Quantitative results.* We quantified the quality of stroke digitisation by analysing the amount of missing strokes and ghost strokes—strokes that were added in error. The analysis was conducted by comparing a scanned worksheet (scanned on the Canon iR-ADV C5035 with 600 dpi colour scan in a lossless format) with the digital counterpart captured on the tablet computer (see Figure 7), where the amount was measured in the number of pixels of which each stroke consisted.

For the missing pixels’ calculation, we manually aligned the scanned worksheet image using the digital counterpart as the baseline. Next, we converted both images to a binary form and performed an absolute image subtraction. The subtracted image was then used to calculate ratios of missing pixels (missing strokes). To obtain the number of ghost strokes, we followed the same procedure, but used the scanned image as a baseline instead.

The results demonstrate a low amount of missing strokes ($\bar{x}=1.374\%\;and\;\sigma =2.248$), as well as a low percentage of ghost strokes ($\bar{x}=1.151\%\;and\;\sigma =1.05$). The *t*-test of unequal variance showed the two values are not significantly different ($t\left(10\right)=0.312,\;p=0.7615,\;Cohen^{\prime }s\;d=0.12,\;conf.\;int.=\left[-0.013,0.0181\right]$). The distribution of ghost strokes (Figure 8b) and missing strokes (Figure 8c) shows that missing strokes have a higher deviation (mostly generated by four participants) compared to ghost strokes (seven participants).

*Qualitative results.* The results in Figure 8d show that a vast majority of participants did not find it hard to draw/sign/write (Q1) using the *Hybrid Stylus* (92%, 92%, and 83% respectively). Figure 8d also shows that the majority of participants were happy with drawing/signing/writing (Q2) (92%, 100%, and 92% respectively). Despite errors detected in quantitative evaluation, these results suggest that the quality of stroke digitisation was still adequate.

The results in Figure 9a show that SUS score is substantially higher than the average score of 68.8 ($\bar{x}=81.4;\;\sigma =12.0$). This suggests that the proposed system offers substantially higher usability compared to an average system. Based on the standard interpretation of the UEQ scale, where values higher than 0.8 present a good score, our *Hybrid Stylus* demonstrated positive results for all dimensions (see Figure 9b). Attractiveness ($\bar{x}=1.833,\;\sigma =1.15$) and Pragmatic Qualities [Perspicuity ($\bar{x}=2.729,\;\sigma =0.35$), Efficiency ($\bar{x}=2.042,\;\sigma =1.11$) and Dependability ($\bar{x}=2.042,\;\sigma =0.60$)] received the highest scores. This is correlated to (i) the short learning curve, (ii) previous experience with tablet computers and styli, and (iii) participants’ familiarity with pencil and paper. Despite good scores for Hedonic Qualities [Simulation ($\bar{x}=1.500,\;\sigma =1.56$), and Novelty ($\bar{x}=1.313,\;\sigma =1.01$)], this result needs to be taken lightly, as the system was not demonstrated or used in a real world scenario/application.

Finally, during informal interviews, participants mentioned that they enjoyed the experience of writing directly on paper and seeing their notes captured on a tablet computer. They also mentioned that usually they do not feel comfortable writing or drawing directly on a tablet screen and they found our approach interesting.

## 5. Practical Applications of the Hybrid Stylus

We designed and developed three use-case prototypes that demonstrate the possibilities of our *Hybrid Stylus*. The application (developed as described in Section 3.2), renders digital counterparts (i.e., the form and the blueprint) as raster graphics on top of which a canvas layer with pen strokes is added. For demonstrative purposes, the prototypes also feature manual paper identification by selecting an appropriate button on the top of the screen. Automatic continuous paper tracking, automatic paper identification, and use of editable digital documents go beyond the scope of the paper.

We focus on novel use-cases, where simultaneous drawing and handwriting on paper and a tablet computer is useful and needed. The use-cases are: form filling, writing sticky notes, and annotating blueprints. They are described in the following text and a video figure that is available as Appendix A.

### 5.1. Form Filling

Forms, such as visa applications, medical diagnoses, or check lists are common in all segments of our society. We show one such form in Figure 10. Users can place the form on the tablet screen and align its position by performing stroke actions over printed target markers. When the position is aligned, the application places a digital counterpart of the physical form exactly below the paper. By doing so we enable users to input the required information synchronously on paper and the tablet computer.

### 5.2. Writing Sticky Notes

Using sticky notes is a popular way to externalise ephemeral information and/or quickly write down reminders. However, users tend to lose, misplace or misshape such notes so information can get lost. Our prototype (Figure 11) enables users to simultaneously generate digital and physical sticky notes. As such, the physical sticky note maintains its original affordances whilst the digital copy makes it more difficult to lose such information.

### 5.3. Annotating Blueprints

At construction sites, architects and builders need to agree on solutions to problems that arise during construction. This usually requires immediate action and happens in the context of a building site by drawing solutions and annotations on paper blueprints. However, these blueprints are held at the construction site, which makes it difficult to synchronise digital versions of the blueprints with the new up-to-date information added on paper. Blueprint annotation prototype (Figure 12) attempts to resolve this problem by enabling the architects and builders to annotate the blueprints by sketching and writing directly on the blueprint whilst the annotations are simultaneously also digitised. After the annotations are made, the hard copy can be given to builders whilst the architects keep the digital counterpart, which can be archived in the cloud.

The same idea can be expanded to any printed document one wishes to annotate. For example, one can annotate a research paper while annotations are directly stored in the digital counterpart. Moreover, to showcase these examples, we used a tablet computer, but any horizontal interactive touchscreen such as a large tabletop or surface computer can be used. When a paper document is placed on such a screen, it merges into the surrounding context and the ecosystem of the underlying digital interface, as presented in Figure 1. In such an ecosystem, the *Hybrid Stylus* provides the benefit of concurrently interacting with both mediums using one tool only.

## 6. Discussion

We organise this discussion around the findings of our experimentation process. Hereafter, we list a series of subsections where we try to address specific issues related to the *Hybrid Stylus*.

### 6.1. Why Is a Dot Better than a Cross?

In the elicitation study, we tried to identify an easy-to-use method for aligning the position of a paper on a tablet display by asking participants to use a pen stroke of their choice. The alignment of the paper position was needed for a fully working prototype. All participants suggested either a cross (66.7%) or a dot (33.3%) stroke action as their first response. Interestingly, the percentage was reversed in Study A, where participants actually needed to use these stroke actions to align the paper position and where the dot stroke action was preferred by 75% of participants. We hypothesise that participants realised they are much faster aligning paper position using the dot stroke action, and thus preferred it over the cross stroke action.

Study A also revealed that participants are quicker and more accurate while performing the dot stroke action. When comparing the positional error (the distance between the centroid on the paper and the screen) between the dot and cross stroke actions at each target marker, the cross stroke action contributed to more errors at both markers but a significant difference was only detected for the target marker 1 (TM1). When comparing the positional error between target markers for each stroke action separately, the results also demonstrate that for each stroke action, participants had bigger positional error at TM1, although the difference is again bigger for the cross stroke action. The biggest error at TM1 can be attributed to our baseline measuring system that logs the paper position only once when the stroke action at TM2 is performed. However, participants have most probably accidentally moved the paper before completing the stroke action at TM2 and the centroids for TM1 moved further apart. The biggest error for cross stroke action can be attributed to its complexity, since the cross stroke action consists of two longer strokes compared to the dot.

Nevertheless, our paper position alignment technique worked well enough for evaluating our *Hybrid Stylus* prototype and to demonstrate three use-cases. However, it also has some drawbacks, since it is not actively tracking paper position. Solutions presented in Section 2.3, their combination, or even novel techniques (see Section 6.4 for our proposal) can be implemented for active tracking and identification.

### 6.2. The Quality of Stroke Digitisation

To the best of our knowledge, no previous accuracy evaluation of pen stroke digitisation for alternative systems such as Anoto or CoScribe digital pens exists. However, we can still conclude that the proposed system offers digitisation of sufficient quality based on the results of Study B: (i) the majority of participants were satisfied with their stroke digitisation quality when writing, signing, or drawing; (ii) the SUS score is high above average; and (iii) the UEQ scores show high pragmatic qualities, and a good overall user experience score.

However, we still observed some shortcomings of our *Hybrid Stylus*. For some users, digitisation errors occurred in the form of missing and ghost strokes. Several reasons might be the cause for this. Some information might have been lost in the process of scanning. However, we scanned the paper at the highest possible dpi in a lossless format. Another reason might be the speed with which participants completed the tasks. However, observations demonstrated that errors occurred even for slower participants. The most probable reason is the way some users held the pen. The digitisation errors occurred when the pen was tilted at a sharp angle. This brought the aluminium shield to a close proximity of the tablet screen, which interfered with the stylus tip detecting its position. We also observed more errors when participants held the stylus without touching the aluminium shield. This happened despite the fact that participants were instructed to hold the pen in a way to also touch the aluminium shield. In the future, alternative shield designs should be explored to mitigate the tilt problem and the problem of not touching the shield. For example, the aluminium shield should be expanded to cover the entire stylus holding area, while the part closer to the screen should be adapted (e.g., cone-shaped). The aluminium could be even integrated in the adapter to mitigate the tilt problem with just parts of it being exposed on the surface (e.g., in a polka dot or ring pattern) where the stylus is held. Whatever the modifications, the ergonomics of the stylus should be considered.

Some errors were also observed with very quick stroke actions, such as signatures. It is not clear if these errors were caused by the modifications that were made to the active stylus or are the limitations of the stylus per-se. Despite the majority of participants in our study being happy with their digitised signatures, this should be explored in more detail within further studies. These could compare digitisation performance (e.g., missing strokes) of our*Hybrid Stylus* tip and other existing hybrid stylus solutions, such as Anoto digital pen.

### 6.3. Lessons Learnt

Through the design process of the hybrid smart pen, several lessons were learnt. For instance, the observations demonstrated that the shield is required for the system to operate as expected, and that its size, shape, and position play a very important role in how well the system performs. In particular, the distance between the extension tip end and the shield is instrumental. This distance was set to nine millimetres (9 mm) in the final prototype and was defined through iterative experimentation with the shield position. Furthermore, the spacing between the extension tip core and the shield also significantly affects the performance. In our system, the diameter of the shield is 9 mm, creating a 3 mm gap between the extension tip core and the shield.

The *Double pen* design (see Figure 2a) makes the stylus heavier, which might affect prolonged usage as well as the pen’s ergonomics. Furthermore, as the stylus uses two tips that are placed at very close proximity, it limits how it can be held and forces the user to maintain the tilt angle. However, it offers the unique flexibility of using any pen or pencil the user has available. In our final design, these shortcomings are overcome as we replace the second pen or pencil by extending the tip of the stylus with an infinite graphite nib (see Figure 2b).

### 6.4. Further Improvements of the Hybrid Stylus

Besides the already mentioned redesign of the aluminium shield, there are several other possibilities to further improve the *Hybrid Stylus*. One example is expanding the functionality of a hybrid stylus with an eraser. Using the right combination of materials, it should be possible to modify the top of the active stylus for simultaneously erasing both content on the paper as well as content on the touchscreen. Another feature to consider would be the design of tips that would be easily changed, supporting multiple colour drawing. Moreover, future studies should experiment with different tip (nib) diameters to explore their effect on usability and stylus performance. One way to identify a particular mounted tip (and its colour) could be conducted through integration of this information within the tip changing process.

The functionality of the stylus could also be enhanced by gaining access to the stylus driver, which would enable us to re-calibrate the modified stylus or even attempt to automatically identify the mounted tip using built-in sensors. Access to the driver would also allow exploration of alternative ghost touch algorithms for the minimisation of stroke errors. Finally, since we modified the stylus, its ergonomic properties have probably been affected. As mentioned earlier, for a market-ready product, ergonomics should be taken into account. We could integrate various sensors to compare its usage to usage of the classic stylus as well as enable extra functionalities [73].

Furthermore, the current version expects the paper to be still after aligning its position. In the current system, the paper is not attached in any way to the tablet computer, which introduces some usability issues, especially in mobile contexts. This issue could be addressed by designing mechanical solutions for attaching paper to the tablet computer with, for example, paper adhesives or clips. Or even better, with an alternative method for actively tracking paper on the tablet screen. Future research could, for example, focus on investigating how this could be achieved by using the underlying capacitive sensing capabilities of a touchscreen. Finally, a robust and user-friendly document identification technique would greatly expand the usefulness of the proposed system, especially if a scalable solution to a large number of documents could be uncovered.

### 6.5. Study Limitations

The studies have a relatively small sample size. However, as the standard deviation within the captured data is small, we believe the results are still valuable. When evaluating stroke digitisation, we only focus on position, ignoring the tilt and pressure functions. In the future, this should be studied in more detail.

## 7. Conclusions

The distinct properties and affordances of paper provide benefits that enabled paper to maintain an important role in the digital age. This is so much so, that pen–paper interaction has been imitated in the digital domain with touchscreens and styli. Despite lacking paper affordances and relying on hardware, software, and power supply, digital documents also possess several advantages that are not available in the physical space, such as editing, copying, and sharing over the internet. It is thus not surprising that a plethora of ideas for merging the two mediums have been proposed, developed, and studied in academia and industry. In this paper, we present the design, fabrication, and evaluation of such an idea – a new prototype called the *Hybrid Stylus* that can write on a physical paper, tablet screen as well as on both surfaces at the same time.

To achieve this, we designed and fabricated the adapter that enables us to attach an infinity pencil tip (nib) made of graphite to an active stylus. The tip acts as a normal pencil and maintains all conductive properties of an active stylus. The *Hybrid Stylus* thus presents a low-cost alternative to handwriting digitisation using modified off-the-shelf hardware. The evaluation through two user studies demonstrated that the system achieves almost perfect digitisation of strokes (98.6% of strokes correctly registered and only 1.2% of ghost strokes), while it maintains excellent user experience of writing with a pencil on the paper (SUS score = 81.4).

This research opens up several directions for future exploration, such as how to expand the functionality of a hybrid stylus (e.g., eraser that can remove content form the paper document and its digital counterpart; replaceable tips to support multi colour drawing and writing on both mediums; introducing novel paper identification methods; active tracking solutions) and various improvement methods for better stroke registration, such as pen tip calibration, and better ghost strokes detection through low level access to stylus and tablet drivers.

## Figures and Tables

**Figure 1 sensors-22-07058-f001:**
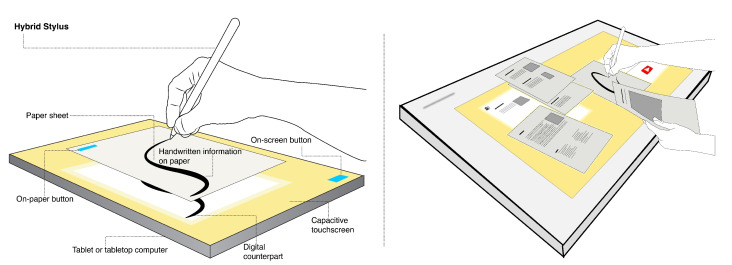
The *Hybrid Stylus* concept. Left: When a paper is placed on a tabletop touchscreen, the user can use the stylus to write on it and simultaneously capture handwritten information in the digital form. The stylus also preserves all capabilities for touchscreen interaction only or can be used solely on paper. Note that the offset between the paper and screen is used only for illustrating digitisation. Right: The vision of using the stylus interchangeably on paper and tabletop computer desk. When a paper document is placed on such a screen, it merges into the surrounding context and the ecosystem of the underlying digital interface.

**Figure 2 sensors-22-07058-f002:**
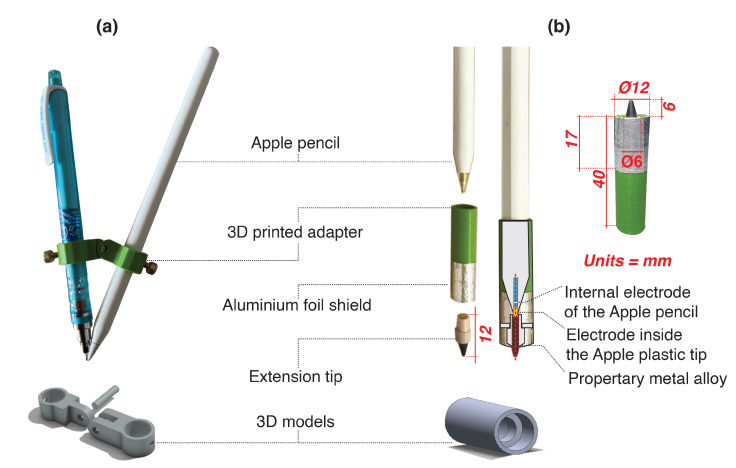
(**a**) *Double pen* adapter design. (**b**) Final design of the *Hybrid Stylus*.

**Figure 3 sensors-22-07058-f003:**
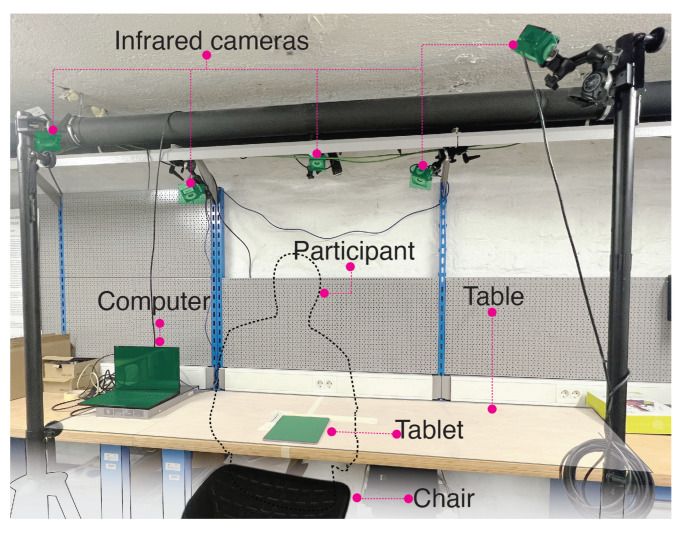
Testing rig used in the three studies.

**Figure 4 sensors-22-07058-f004:**
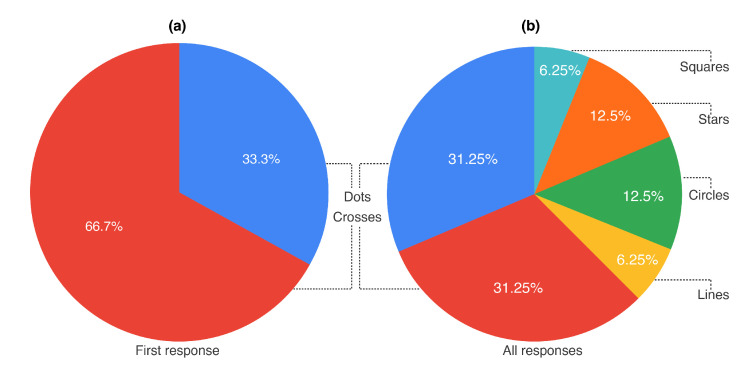
(**a**) Result of the first response to the question: What gesture would you most likely make with a stylus to instruct the system of the exact location of the paper on a tablet screen? (**b**) Aggregated results for all three responses.

**Figure 5 sensors-22-07058-f005:**
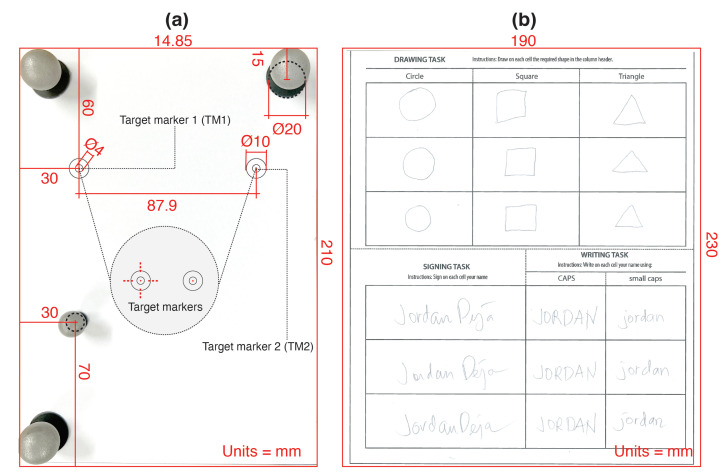
(**a**) A5 size paper used in Study A with two target markers (TM1 & TM2). Participants had to perform a stroke action over these markers to align paper position. One paper had both target markers presented as a cross and the other as a dot. (**b**) Worksheet used in Study B with drawing, signing and writing tasks to evaluate the usability of the *Hybrid Stylus* as a regular pen.

**Figure 6 sensors-22-07058-f006:**
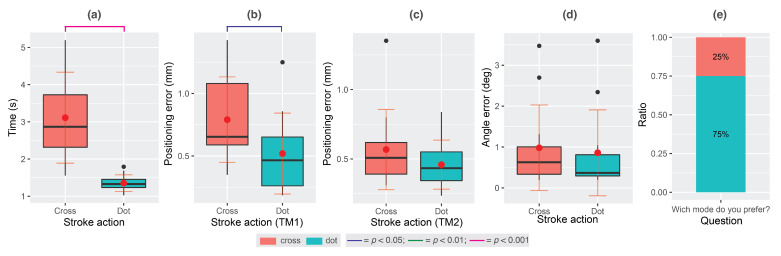
(**a**) Task completion time. (**b**) Position error for target marker 1 (TM1). (**c**) Position error for target marker 2 (TM2). (**d**) Angle error. (**e**) Preferred stroke action.

**Figure 7 sensors-22-07058-f007:**
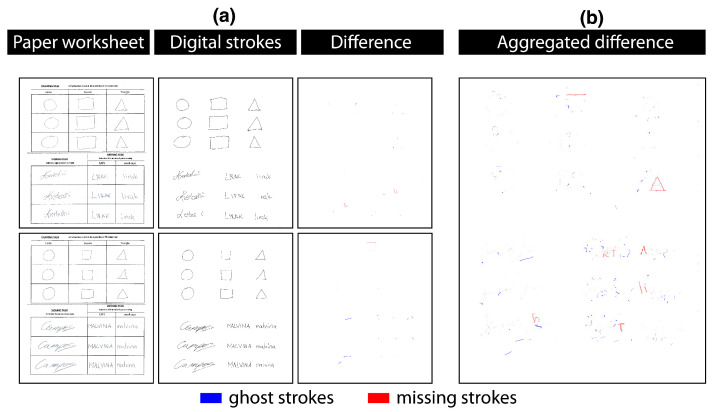
Visual representation of the procedure for image analysis. (**a**) Two examples of paper worksheets and their digital counterparts from two participants, together with differences between the two mediums. (**b**) Aggregated representation of missing and added strokes of all participants.

**Figure 8 sensors-22-07058-f008:**
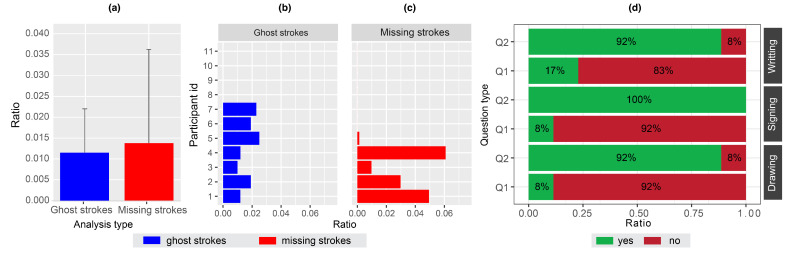
(**a**) Overall quantitative analysis for added and missing strokes. (**b**) Quantitative analysis of added strokes by participants. (**c**) Quantitative analysis of missing strokes by user. (**d**) User preference (Q1: Was it hard to write, sign or draw? Q2: Are you satisfied with the digital counterpart of your writing, signature, and drawing?).

**Figure 9 sensors-22-07058-f009:**
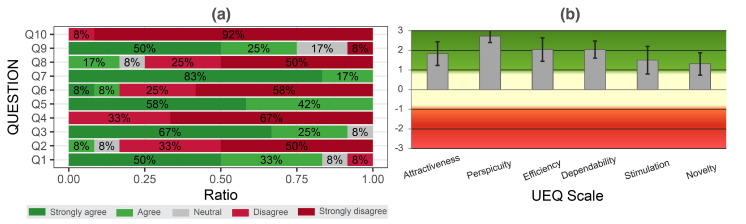
(**a**) Graphical representations of the System Usability Scale (SUS) Questionnaire (SUS: https://www.usability.gov/how-to-and-tools/methods/system-usability-scale.html (accessed on 31 August 2022)). (**b**) Overall results of the User Experience Questionnaire UEQ (long version) (UEQ: https://www.ueq-online.org/ (accessed on 31 August 2022)).

**Figure 10 sensors-22-07058-f010:**
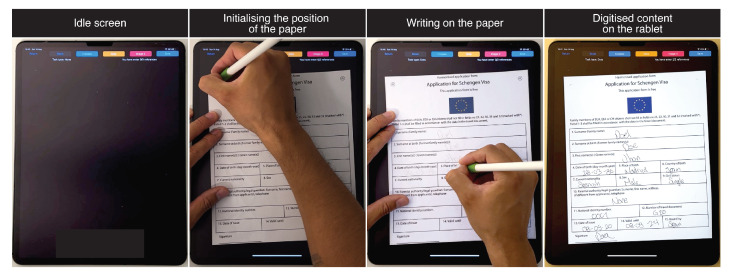
Form filling use-case.

**Figure 11 sensors-22-07058-f011:**
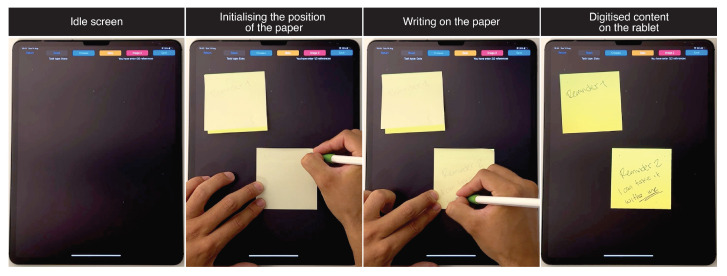
Sticky notes use-case.

**Figure 12 sensors-22-07058-f012:**
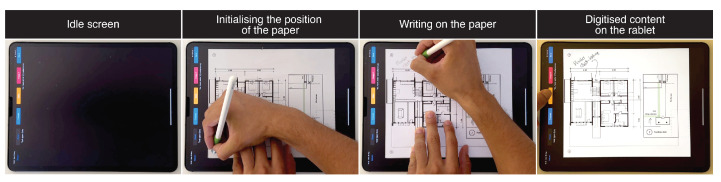
Blueprints annotation use-case.

**Table 1 sensors-22-07058-t001:** The *Hybrid Stylus* keeps all features of an Apple Pencil but one—the pressure sensitivity, which maintains a constant force in our device.

Interaction Technique	Input Method		
	*Finger*	*Apple Pencil*	*Hybrid Stylus*
*touch*	Yes	Yes	Yes
*presses*	Yes	Yes	Yes
*gestures*	Yes	Yes	Yes
*3D force*	Yes	Yes	Yes
*double tap*	No	Yes	Yes

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
