# Peer review of "The Hybrid Stylus: A Multi-Surface Active Stylus for Interacting with and Handwriting on Paper, Tabletop Display or Both"

_sensors, 2022, doi:10.3390/s22187058_

Round 1

Reviewer 1 Report

The authors have proposed the design, fabrication, and evaluation of a hybrid stylus and the proposed two-point technique for initializing paper position on the underlying tablet computer. The contribution is OK to me. The paper is quite well-written, although some typos are still found. It should be revised. The results of quantitative analysis of missing strokes by users are quite promising to me. But the position error on TM1 should be explained more clearly in the case of the cross gesture. However, some literature reviews for hand gesture classification should be discussed more clearly. For example, several recent papers for hand gesture classification should be added and discussed, such as SketchADoodle: Touch-surface Multi-stroke Gesture Handling by Bézier Curves. Proc. ACM Hum. Comput. Interact. 4(EICS): 87:1-87:30 (2020), Deep Learning Based Gesture Classification for Hand Physical Therapy Interactive Program. HCI (18) 2020: 349-358 and User-Independent Hand Gesture Recognition Classification Models Using Sensor Fusion. Sensors 22(4): 1321 (2022). Also, some following literature reviews for human factor studies should be included, such as Human behavior in image-based Road Health Inspection Systems despite the emerging AutoML. J. Big Data 9(1): 96 (2022) and Assistive Handlebar Based on Tactile Sensors: Control Inputs and Human Factors. Sensors 18(8): 2471 (2018). In conclusion, the paper could be accepted after proper revision. If not, I am afraid to reject this paper.

Author Response

Dear reviewe,

Our reply is available in the attached pdf.

Best,

Authors

Reviewer 2 Report

The authors present a method to augment an Apple Pencil to support sketching on a physical paper as well as on tablet screens. They propose to use conductive graphite to connect classic stylus as their design and conduct user studies to validate their design. While I agree that the solution is simple and the proposed stylus is beneficial, there are still many problems about this paper: 

- I cannot get the motivation of the user study. For user study A, I don't think the results are useful and related to support the designed stylus. It simply compares dot and cross strokes. If this work aims to study which stroke pattern is good, I can understand the design of this user study. As this work claims its contribution as the 'hybrid stylus', I don't think user study A is necessary. Instead, I expect to see the comparison with other solutions that may also support 'hybrid writing', so that they can use the results to validate the benefits of the design. Or they may compare it with classic Apple pen, so that they may conclude the added features do not sacrifice the user experience. Unfortunately, I cannot either find these results in user study B.

- In the introduction section, the authors mentioned that they explored the research question 'How to track paper on the underlying tablet computer ....'. However, I cannot find the detailed answer to this question in the following sections. They simply said it 'goes beyond the scope of the paper', which is disappointing.  

- How do you measure the angle error in section 4.2? If a user draws a dot, it is not possible to extract the angle error as the dot is isotropic.  

- In fig.8(b,c), why several participants do not have ghosted or missing strokes?

- a missing reference 'DodecaPen: Accurate 6DoF Tracking of a Passive Stylus. ACM UIST 2017'

- of-the-shelve --> off-the-shelve, Finlay --> finally, duplicated 'the' (line 284), trough--> through (line 223)

Author Response

Dear reviewer,

Our reply is available in the attached pdf.

Best,

Authors

Reviewer 3 Report

In this work, for “The Hybrid Stylus: a multi-surface-active stylus for interacting with and handwriting on paper, tabletop display or both”. I think this work is scientifically significant as well as application-oriented. Therefore, this manuscript can be considered for publication after minor revision. Please find my comments below.

1.      Please explain the novelty of this hybrid Stylus in the introduction section; we already know that there are so many technologies available with the advanced technology with more accuracy. As you introduce the pencil graphite nib in the apple pen then how will it be cost-effective and novel.?

2.      In Figure 7.  theVisual representation of the procedure for image analysis” shows that there are so many errors that digital and paper worksheets are not the same. For example, if the “sign” is not the same then how will you say that its “Hybrid Stylus” is the futurist model?

3.      Author should also add the table of “comparison of touch approaches along various dimensions that influence interaction”.

4.      Author adds the “parameters used in the comparison of the unintended touch algorithms given current and future functionality”.

5.      Please also explain the resulting effect sizes for the Hover Cursor, Hand Resting, and Nib Diameter conditions.

6.      Author should also check the participants resting their forearm on the bottom bezel and bottom of the touch screen that generates touch input.

7.      The conclusion should be very much précised and focused and rephrased this line for better connectivity “And because touchscreens imitating paper interaction with stylus pens are also pervasive the two mediums would benefit form merging”.

Author Response

(The authors gave the same response as above.)

Round 2

Reviewer 2 Report

My concerns are addressed in the revision. Although I still expect comparisons with other hybrid stylus, the manuscript is ready to go given the time limits. 

Reviewer 3 Report

Thank you for your response.